# Clinical Results of the Implementation of a Breast Milk Bank in Premature Infants (under 37 Weeks) at the Hospital Universitario del Valle 2018–2020

**DOI:** 10.3390/nu13072187

**Published:** 2021-06-25

**Authors:** Javier Torres-Muñoz, Carlos Alberto Jimenez-Fernandez, Jennifer Murillo-Alvarado, Sofia Torres-Figueroa, Juan Pablo Castro

**Affiliations:** 1Department of Pediatrics, Health Faculty, School of Medicine, Universidad del Valle, Cali 100-00, Colombia; carlos.jimenez.fernandez@correounivalle.edu.co (C.A.J.-F.); jennifer.murillo@correounivalle.edu.co (J.M.-A.); 2Health Sciences Faculty, School of Medicine, Universidad ICESI, Cali 122-135, Colombia; sofitorres99@hotmail.com (S.T.-F.); juanpcastro2698@outlook.es (J.P.C.)

**Keywords:** human milk banks, enteral nutrition, breast milk, enterocolitis, sepsis

## Abstract

Breast milk is widely recognized as the best source of nutrition for both full term and premature babies. We aimed to identify clinical results of the implementation of a breast milk bank for premature infants under 37 weeks in a level III hospital. 722 neonates under 37 weeks, hospitalized in the Neonatal intensive care unit (ICU), who received human breast milk from the institution’s milk bank 57% (*n* = 412) vs. mixed or artificial 32% (*n* = 229), at day 7 of life. An exploratory data analysis was carried out. Measures of central tendency and dispersion were used, strength of association of odds ratio (OR) and its confidence intervals (95% confidence interval (CI)). 88.5% had already received human milk before day 7 of life. Those who received human milk, due to their clinical condition, had 4 times a greater chance of being intubated (OR 4.05; 95% CI 1.80–9.11). Starting before day 7 of life decreases the opportunity to develop necrotizing enterocolitis by 82% (adjusted odds ratio (ORa) 0.18; 95% CI 0.03–0.97), intraventricular hemorrhage by 85% (ORa 0.15; 95% CI 0.06–0.45) and sepsis by 77% (ORa 0.23; 95% CI 0.15–0.33). Receiving human milk reduces the probability of complications related to prematurity, evidencing the importance that breast milk banks play in clinical practice.

## 1. Introduction

Breast milk (BM) is widely recognized as the best source of nutrition for healthy full-term babies, besides offering many benefits for those born prematurely. Moreover, some of these benefits can also be observed when donor milk is used instead of the mother´s. Neonatal ICUs should provide support and education to mothers, promoting breastfeeding in order to reduce infection rates, necrotizing enterocolitis (NEC) and mortality, as well as better long-term neurocognitive and cardiovascular outcomes [1,2,3] thus impacting mortality and overall costs in health systems. When BM is not available, the recommended alternative (with sufficient evidence for preterm infants and patients hospitalized in ICUs) is the use of donated and pasteurized human milk (DM) obtained from selected healthy donors, and derived from Milk Banks. According to the World Health Organization, prematurity is the leading global cause of mortality in children under five years of age [4]. In fact, in low-income countries half of the babies born at 32 weeks die because of a lack of simple and cost-effective care, not providing breastfeeding support, no basic infection-fighting care and respiratory problems [4]. Therefore, BM Banks represent a strategy for neonatal and infant survival. With the use of promotion, protection and support of breastfeeding, in addition to the processing, quality control and supply of human milk to hospitalized neonates, there is a guarantee of food and nutritional security for premature infants. With its adequate use, BM banks contribute to the reduction of neonatal and infant malnutrition, morbidity and mortality.

Human milk banks are increasing in the world; in Europe there are around 210 new banks and 17 in the process of opening, which are being established with the support of the Association of European Milk Banks. In the United States, the routinary use of both human and donor milk in ICUs increased from 21.1% in 2007 to 30.8% in 2011 [5]. Right now in Colombia there are 9 human milk banks distributed throughout the country. The institution where the present study was carried out began its construction of the Milk Bank in 2014, with the support of the Municipal Health Secretariat and the goal to be recognized as a Specialized Education Center to Promote and Encourage Exclusive Breastfeeding. Therefore, there is a need to carry out research work, as the results will be important inputs to establish management guidelines in the institutions that desire a Milk Bank, especially in Latin America. The objective of this study was to analyze the clinical results of the implementation of a human milk bank for premature infants under 37 weeks in a level III hospital (Hospital Universitario del Valle, Cali, Colombia).

## 2. Materials and Methods

### 2.1. Study Setting

This study was done with the approval of the Ethics Committee of the Hospital Universitario del Valle (HUV) and the Universidad del Valle and was developed in the CIRENA neonatal intensive care unit at the Hospital Universitario del Valle, which has a human milk bank. The study design is descriptive, observational, cross-sectional, of premature newborns during 2018 and 2020, under 37 weeks, hospitalized in the Neonatal ICU who received human breast milk from the institution’s milk bank and whom information was collected from the clinical record and from the institution’s human milk bank forms.

### 2.2. Object Population of Study

Preterm newborns with a gestational age less than or equal to 37 weeks who are hospitalized in the neonatal ICU, and who receive breast milk or milk from the human milk bank. Patients with major congenital malformations or metabolic diseases were excluded from the study.

### 2.3. Sample Size

All premature newborns hospitalized in the Neonatal ICU during the years 2018 to 2020 were included. During this period, 17,007 were born in this institution, corresponding to 5102 premature infants, information was collected from 722 participants who were admitted to the intensive care.

### 2.4. Statistical Analysis

An exploratory analysis of the data was carried out by means of a univariate and bivariate analysis where the distributions of continuous variables and frequency distributions in qualitative variables, loss of data and the consistency of the information were explored. Comparisons between interest groups were made using statistical chi-square tests and Fischer’s exact test as appropriate. Quantitative variables were compared using the Wilcoxon test for paired data. The strength of association of the OR and its confidence intervals (95% CI), crude and adjusted, was determined by means of a logistic regression model and a value of *p* less than 0.05 was considered significant. Data analysis and processing was performed with the STATA version 2014 program.

## 3. Results

### 3.1. General Characteristics

The sample includes all babies under 37 weeks admitted to the neonatal intensive care unit.

723 preterm infants under 37 weeks were admitted to the study (Table 1). Of these, one patient was excluded due to insufficient data (Figure 1). So, 722 babies were analyzed (Figure 1). In 99% of the Cases BM was collected, and in 98% the babies were fed milk from their own mother. 50% of the premature babies were younger than 33 weeks. 57% (*n* = 412) were fed with only human milk at day 7 of life and 32% received mixed or artificial milk (*n* = 229) (volume of 150 mL/kg/d of enteral feeding). 1.67% (12 cases) developed stage II or III NEC according to the modified Bell criteria [6]. On the third day of life, 70% already received enteral feeding with human milk, and 52% reached a 100% (150 mL/kg/day) of enteral feeding at day 7.

### 3.2. Comparison by Type of Milk that the Newborn Receives at 7 Days of Age and the Number of Days It Takes to Reach Full Breastfeeding

When evaluating the results depending on the type of milk received at day 7 of life (Table 2), it was observed that babies fed with human milk were 4 times more likely to enter critical conditions with need of mechanical ventilation, compared to those with mixed or artificial milk (OR 4.05; 95% CI 1.80–9.11).

Late preterm have a greater chance of receiving breast milk (OR 2.89; 95% CI 2.06–4.05 *p* < 0.001), achieving complete feeding (defined as 150 mL/kg/day) before 7 days was significantly faster in the human milk group (with 18 times greater chance of achieving it) compared to those who received artificial milk or mixed (OR 18.88; 95% CI 12.37–28.81; *p* < 0.001). Babies fed with human milk had a higher chance of adequate prenatal control (OR 2.45; 95% CI 1.67–3.57) and fewer days of hospitalization (OR 23.30; 95% CI 9.38–57.86).

When analysing crude OR, patients who reached 100% of feeding with human milk before 7 days (Table 3) had a significantly lower probability of necrotizing enterocolitis (OR 0.16; 95% CI 0.03–0.79), bronchopulmonary dysplasia (OR 0.04; 95% CI 0.00–0.30), intraventricular hemorrhage (0.12 95%; CI 0.04–0.32), retinopathy (OR 0.15; 95% CI 0.03–0.69) and late sepsis defined as newborns who had a positive blood culture after 48 h of hospitalization (OR 0.23; 95% CI 0.16–0.32), compared to those who reached it after 8 days. 

By adjusting the model with the significant variables: type of milk received at 7 days of birth (maternal vs. artificial or mixed (Table 4), it was identified that having adequate prenatal controls increases the opportunity of receiving human milk (ORa 1.95; 95% CI 1.26–3.01) and lowers the chance of late sepsis by 37% (ORa 0.63; 95% CI 0.43–0.93).

Late preterm infants are significantly more probability to receive breast milk, this variable was not included in the final model since it is considered a confounder, as it is the group with the lowest probability of enterocolitis. With adjusted OR, when comparing the time (in days) for reaching feeding with 100% human milk (7 or fewer days vs. 8 or more days) it was found that starting before day 7 of life (Table 5) decreases the opportunity to develop necrotizing enterocolitis by 82% (ORa 0.18; 95% CI 0.03–0.97), intraventricular hemorrhage by 85% (ORa 0.15; 95% CI 0.05–0.45) and late sepsis by 77% (ORa 0.23; 95% CI 0.15–0.33).

The weight (Table 6) of admission in babies fed with only breast milk was significantly higher than those who received artificial or mixed milk with a median of 1705 g vs. 1545 (*p* < 0.01). This was not maintained at 7 day and at discharge, but it showed no significant differences (weight 1697 vs. 1570: *p* = 0.06 and 1910 vs. 1930: *p* = 0.67).

## 4. Discussion

The number of premature babies continues to increase throughout the world, especially in low and middle-income countries. It is necessary to implement practices like BM banks, in order to reduce their complications and promote optimal nutrition and development. Necrotizing enterocolitis is the most serious and frequent intestinal complication in premature infants, especially in those with very low birth weight. According to various observational studies, its incidence fluctuates between 2% and 20% [5,7].

The objective of this study was to analyze the clinical results of the implementation of a human milk bank for premature infants under 37 weeks in a level III hospital (Hospital Universitario del Valle, Cali, Colombia). Out of 1569 admissions in 2018 there were 12 patients who developed stage NEC Bell II or III. This is much lower than the 2008 rates of the same institution in which 1555 were admitted and the frequency of NEC corresponded to 32 cases [8]. There is a significant difference (*p* = 0.001), showing the benefits of the implementation of a human milk bank.

The results show how the early start of enteral feeding with BM before the 7th day of life reduces the chance of developing necrotizing enterocolitis by 93% (OR 0.07; 95% CI 0.01–1.17) compared with those who received mixed feeding or only artificial milk, non-significant result. Thus, evidencing the clear benefit of BM in this population. Other studies that demonstrate the benefits of early initiation of human milk and the reduction of necrotizing enterocolitis up to 50% have been published as reported by Dr. Cristofalo [9]. In systematic reviews and meta-analysis evaluating human milk vs. artificial milk (including 1472 babies from 43 observational studies), the reduction in necrotizing enterocolitis was significant (RR 0.51; 95% CI 0.35, 0.76, *n* = 3783), as well as late-onset sepsis and severe retinopathy [10]. In a systematic review and meta-analysis in which 32 publications were evaluated, babies who received human milk vs. artificial milk had a 38% reduced risk of developing necrotizing enterocolitis (RR = 0.62; 95% CI 0.42–0.93). In observational studies comparing babies fed with human vs. mixed milk, human milk showed a protective effect for necrotizing enterocolitis of 36% (RR = 0.74; 95% CI 0.63–0.91) [11]. 

It was evidenced that those who receive human milk before day 7 of life, have a 37% lower chance of developing late sepsis (ORa 0.63; 95%CI 0.43–0.93). These results are similar to those found by Van Gysel, Dan L et al. [12], which show that human milk contains bioactive substances with bactericidal activity, inhibiting the growth of Escherichia coli, Staphylococcus aureus and Candida sp. In addition, pasteurization prevents the potential risk of transmission of pathogens to premature babies by donors. In a Spanish narrative review [13] on breast milk banking and personalized nutrition in the NICU, there was a decreased late-onset sepsis from 14.7 cases/1000 days to 9.5 cases/1000 days of central lines. It also reduced necrotizing enterocolitis in premature infants of less than 32 weeks from 10.9% (12/110) to 2.4% (2/84). These results support the claim that human milk is not just a nutrient, but also an important therapy for premature and ill babies. 

When comparing the time in days (≤7 days vs. ≥8 days) in which feeding with 100% human milk is reached (150 mL/kg/day), it was found that achieving 100% enteral feeding before day 7 of life, decreased the chance of developing necrotizing enterocolitis by 82% (ORa 0.18; 95% CI 0.03–0.97); intraventricular hemorrhage by 85% (ORa 0.15; 95% CI 0.05–0.45); and sepsis by 77% (ORa 0.23; 95% CI 0.15–0.33). The results are comparable to those obtained in an observational study in a university hospital in Korea [14], in which they report a frequency of enterocolitis of 0 cases in the group fed with breast milk vs. 9.3% for those who received artificial milk, and late-onset sepsis was 1 case (2.8%) vs. 17 (31.5%) respectively, with a protection of breast milk of 94% (OR 0.06; 95% CI 0.01–0.49). Complete enteral feeding at 130 mL/kg/day was reached in a period of 29.6 ± 12.0 days in the group that received human milk from the bank vs. 52.2 ± 17.6 days in the group fed with artificial milk, showing significant differences between both groups (*p* = 0.001). In the present study, the group that received human milk had 18 times a greater probability of achieving complete feeding of 150 mL/kg/day sooner, when compared to those who received artificial or mixed milk (OR 18.88; 95% CI 12.37–28.81 *p* < 0.001).

Studies have evaluated how early onset and amount of milk received has positive effects on these babies. This was reported by Meinzen et al. [15], as increased BM intake in the first 14 days of life in babies with very low birth weight was associated with a longer enterocolitis-free survival time. For every 100 mL/kg increase of human milk there was a longer ECN free time, indicating a dose-response relationship (HR 0.87; 95% CI 0.77, 0.97). Another investigation [16] that evaluated the relationship between the amount of human milk administered and necrotizing enterocolitis and sepsis, concluded that those who receive at least 50% of human milk in the first 14 days of life have a lower probability of necrotizing enterocolitis (OR = 0.17; 95% CI 0.04–0.68; *p* = 0.01).

In this study, when comparing the two forms of milk: human vs. artificial or mixed, there was a greater chance of hospitalization for more than 7 days in the group with artificial or mixed milk (OR 23.30; 95% CI 9.38–57–86). The latter also had a lower number of prenatal controls (defined as less than 3) (OR 2.45; 95% CI 1.67–3.57). 

Furthermore, although the weight at admission for babies fed with only breast milk was significantly higher than those who received artificial or mixed milk, with a median of 1705 g vs. 1545 (*p* < 0.001), this did not remain significant at day 7 and at discharge (weight 1697 vs. 1570; *p* = 0.06 and 1910 vs. 1930 *p* = 0.67 respectively). There was a higher weight gain in those who received artificial or mixed milk. In a systematic review of 12 randomized clinical trials or quasi-randomized studies [17] that compared feeding with artificial milk vs. donor breast milk in a 1879 very low-birth-weight premature infants, the group who received artificial milk had a higher rate of weight gain, but also higher risk of necrotizing enterocolitis. In the long term, it has been found that human milk from the mother or a bank is more suitable for premature infants, as they present risk impaired neurodevelopment, microbiome, chronic disease and other problems associated with prematurity [18,19,20].

Mothers with education up to primary school represented 32.55% of the sample, and 24% of them did not have a partner. 98% of infants received milk from their own mother stored at the milk bank, 88.5% received human milk before 7 days of age and 52% received it exclusively. Those who received human milk due to their clinical status had a 4 times greater chance of being intubated (OR 4.05; 95% CI 1.80–9.11) compared to those who received mixed or artificial milk.

Limitations of the study include: its small sample size and lack of assessment of neurodevelopmental outcomes or cognitive functions after discharge from the neonatal intensive care unit. The study was carried out in a single center, as our hospital is the only center in the region that has a human milk bank. This may favor a selection bias, a possibility that is known in all observational studies and emphasizes the importance of randomized trials to evaluate forms of feeding in preterm infants.

Strengths of the study include: information for members of neonatal intensive care units on the importance of human milk (donors or own mother) in reducing necrotizing enterocolitis, particularly when it is administered early in the life of these babies. The mothers of these premature babies are also encouraged to extract their milk when their baby is hospitalized. Further investigation is required on the factors that determine the success of feeding with mother’s or donor’s milk, and on the importance of human milk banks for premature babies who are hospitalized in neonatal intensive care units.

## 5. Conclusions

Institutions that have a breast milk bank and are able to preserve it and promote donations, facilitate a health improvement for newborns who are hospitalized in intensive care units.

Breast milk banks are an absolute must, especially for premature babies. Unfortunately, these are not present in every country in the world. The collection and processing of breast milk is established by guidelines. Globally, there is growing interest in increasing milk banks through awareness campaigns to donate milk.

Human donor milk, especially from the mother, not only represents the best nutrition for babies, it is also one of the most important therapies used in neonatal intensive care, protecting the babies against various diseases when used early. A better knowledge of its biological characteristics will facilitate innovative processes for medical use.

## Figures and Tables

**Figure 1 nutrients-13-02187-f001:**
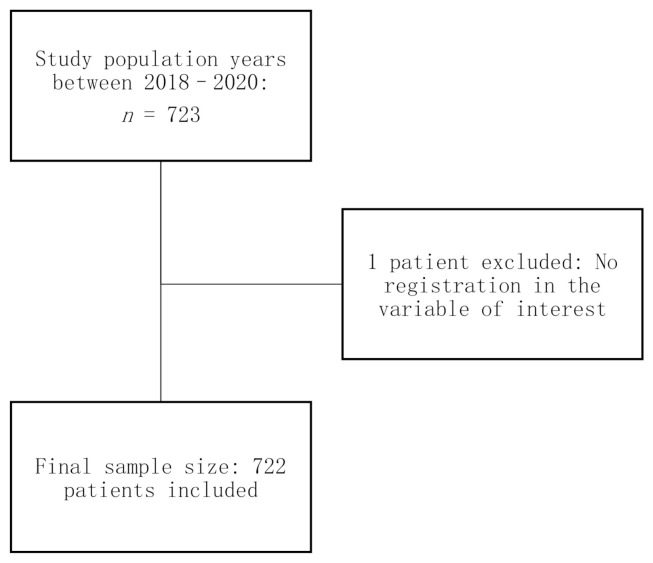
Total patients evaluated during the study; 2018–2020, children under 37 weeks hospitalized in the neonatal intensive care unit.

**Table 1 nutrients-13-02187-t001:** General characteristics of premature infants under 37 weeks hospitalized in the neonatal intensive care unit admitted to the study.

Variable		Frequency (*n* = 722)	%
Maternal variables			
Age	<19 years	103	14.27
	19–35	548	75.90
	>35	67	9.28
Scholarship	No schooling	twenty	2.77
	Primary	215	29.78
	Secondary	313	43.35
	Further education	122	16.90
With partner	Yes	548	75.9
Origin	Cali	434	60.11
Donor type	Homologous Donor *	709	98.2
	Heterologous Donor **	13	1.8
Institutional Collection	HUV ****	714	98.89
Received beastfeeding education	Yes	241	33.38
Pregnancy	First	304	42.11
Prenatal controls	Suitable ***	536	74.24
Maternal pathologies	None	156	21.61
	Hypertensive illness	261	36.15
	Infectious disease	227	31.44
	Other	60	8.31
Newborn variables			
Sex	Female	358	49.58
	Male	363	50.28
Gestational age	Equal to or less than 33 weeks	366	50.69
	Between 34–37	356	49.31
Discharged	Alive	685	94.88
	Dead	36	4.99
Way of birth	Vaginal	363	50.28
Intubated	Yes	685	94.88
Hospitalized days	Less than 7	158	21.88
	7 or more	564	78.12
Necrotizing enterocolitis	Stage II	9	1.25
	Stage III	3	0.42
Intraventricular hemorrhage	Stage I	28	3.88
	Stage II	4	0.55
	Stage III	7	0.97
Bronchopulmonary dysplasia	Mild	7	0.97
	Moderate	13	1.8
	Severe	8	1.11
Retinopathy	Stage I	11	1.52
	Stage II	2	0.28
Late Sepsis	Yes	275	38.09
Breastfeeding variables			
Start day of consumption of Breast milk	Three days	503	69.67
	Between 4 to 7 days	136	18.84
	8 and more days	70	9.7
100% Breast Milk Days	Three days	230	31.86
	Between 4 to 7 days	154	21.33
	8 and more days	304	42.11
Type of milk received at 7 days old	Breastmilk	412	57.06
	Artificial milk	5	0.69
	Mixed	224	31.02

* Homologous donor corresponds to milk from his own mother. ** Heterologous donor corresponds to milk from another mother. *** Adequate prenatal control defined greater than 3 controls. **** Hospital Universitario del Valle.

**Table 2 nutrients-13-02187-t002:** Type of milk administered (human vs. mixed or artificial) at day 7 of life, to premature infants under 37 weeks hospitalized in the neonatal intensive care unit.

Variable		Type of Milk Received at 7 Days Old		*p*-Value	OR (95% CI)
		Maternal *n* = 412 (%)	Other * *n* = 229 (%)		
Maternal age	<19 years	43 (10.49)	42 (18.50)	0.005	0.51 (0.32–0.81)
	19 or more	367 (89.51)	185 (81.50)		
Origin	Cali	264 (64.08)	132 (58.41)	0.158	1.27 (0.91–1.77)
	Outside of Cali	148 (35.92)	94 (41.59)		
Education in breastfeeding	Received	148 (36.54)	70 (31.25)	0.182	1.26 (0.89–1.79)
Pregnancy	First	171 (41.50)	93 (40.61)	0.826	1.03 (0.74–1.44)
Gestational age	Equal to or less than 33 weeks	165 (40.05)	151 (65.94)	<0.001	2.89 (2.06–4.05)
	Between 34–37	247 (59.95)	78 (34.06)		
Adequate prenatal control	Suitable *	334 (82.67)	148 (66.07)	<0.001	2.45 (1.67–3.57)
Intubation	If required	403 (97.82)	210 (91.70)	0.001	4.05 (1.80–9.11)
Way of birth	Vaginal	219 (53.16)	104 (45.61)	0.068	1.35 (0.97–1.87)
Days of hospitalization	Less than 7	141 (34.22)	5 (2.18)	<0.001	23.30 (9.38–57.86)
Necrotizing enterocolitis	Yes	1 (0.25)	4 (1.87)	0.07	0.13 (0.01–1.17)
Bronchopulmonary dysplasia	Yes	6 (1.48)	10 (4.69)	0.023	0.30 (0.10–0.84)
Intraventricular hemorrhage	Yes	12 (2.96)	15 (7.04)	0.022	0.40 (0.18–0.87)
Retinopathy	Yes	5 (1.23)	6 (2.82)	0.168	0.43 (0.12–1.42)
Late sepsis	Yes	121 (29.95)	108 (50.70)	<0.001	0.41 (0.29–0.58)
100% milk	before the 7th day	331 (80.93)	40 (18.35)	<0.001	18.88 (12.37–28.81)

* Adequate prenatal control defined greater than 3 controls. Bivariate analysis comparing feeding with human vs. mixed or artificial milk. The crude ORs are shown, using logistic regression. Another corresponds to artificial or mixed milk.

**Table 3 nutrients-13-02187-t003:** Time to reach 100% human milk feeding defined as before 7 days or more than 8 days, in premature infants under 37 weeks hospitalized in the intensive care unit.

Variable		Starting Days 100% Breast Milk		*p*-Value	OR (95% CI)
		7 or Less *n* = 384 (%)	8 or More *n* = 304 (%)		
Maternal age	<19 years	37 (9.66)	57 (18.94)	0.001	0.45 (0.29–0.71)
	19 or more	346 (90.34)	244 (81.06)		
Origin	Cali	250 (65.10)	164 (54.49)	0.005	1.55 (1.14–2.12)
	Outside of Cali	134 (34.90)	137 (45.51)		
Education in breastfeeding	Received	137 (36.34)	91 (30.23)	0.095	1.31 (0.95–1.82)
Pregnancy	First	162 (42.19)	125 (41.12)	0.778	1.04 (0.77–1.41)
Adequate prenatal control *	Suitable *	308 (81.05)	211 (71.53)	0.004	1.70 (1.18–2.44)
Intubation	If required	367 (95.57)	291 (95.72)	0.923	0.96 (0.46–2.01)
Gestational age	Equal to or less than 33 weeks	129 (33.59)	215 (70.72)	<0.001	4.77 (3.44–6.61)
	Between 34–37	255 (66.41)	89 (29.28)		
Way of birth	Vaginal	217 (56.66)	133 (43.75)	0.001	1.68 (1.24–2.27)
Days of hospitalization	Less than 7	150 (39.06)	0 (0.00)	1	1
	7 or more	234 (60.94)	304 (100.00)		
Necrotizing enterocolitis	Yes	2 (0.54)	9 (3.09)	0.024	0.16 (0.03–0.79)
Dysplasia B **	Yes	1 (0.27)	18 (6.16)	0.002	0.04 (0.01–0.31)
Intraventricular Hemorrhage ***	Yes	5 (1.34)	29 (9.93)	<0.001	0.12 (0.04–0.32)
Retinopathy ****	Yes	2 (0.54)	10 (3.42)	0.016	0.15 (0.03–0.69)
Late sepsis	Yes	89 (23.99)	168 (57.73)	<0.001	0.23 (0.16–0.32)

* Adequate prenatal mocontrol defined greater than 3 controls. ** Bronchopulmonary dysplasia defined as oxygen at 28 days of age or 36 weeks of corrected chronological age. *** Intraventricular hemorrhage of any stage. **** Retinopathy of prematurity any stage.

**Table 4 nutrients-13-02187-t004:** Model with the significant variables according to the type of milk (maternal vs. artificial or mixed) received by the premature infants less than 37 weeks of gestation hospitalized in the neonatal intensive care unit, at 7 days of birth.

Variable		OR Crude (95% CI)	*p*-Value	Adjusted OR * (95% CI)	*p*-Value
Maternal age	<19 years	0.51 (0.32–0.81)	0.005	0.69 (0.41–1.18)	0.179
	19 or more				
Gestational age	Equal to or less than 33 weeks	2.89 (2.06–4.05)	<0.001	1.55 (1.05–2.30)	0.027
Adequate prenatal control	Suitable **	2.45 (1.67–3.57)	<0.001	1.95 (1.26–3.01)	0.003
Intubation	If required	4.05 (1.80–9.11)	0.001	1.72 (0.42–6.97)	0.441
Dysplasia B	Yes	0.30 (0.10–0.84)	0.023	0.94 (0.31–2.88)	0.926
Intraventricular hemorrhage	Yes	0.40 (0.18–0.87)	0.022	0.84 (0.36–1.93)	0.686
Late sepsis	Yes	0.41 (0.29–0.58)	<0.001	0.63 (0.43–0.93)	0.023

* Multivariate analysis with adjusted OR using logistic regression. ** Adequate prenatal control defined greater than 3 controls.

**Table 5 nutrients-13-02187-t005:** Time in days that starts feeding l take human to 100% (7 or less days vs. 8 or more days) in 37 weeks premature hospitalized in the intensive care unit neonatal.

Variable		OR* Crude (95% CI)	*p*-Value	Adjusted OR ** (95% CI)	*p*-Value
Maternal age	<19 years	0.45 (0.29–0.71)	0.001	0.37 (0.22–0.64)	<0.001
	19 or more				
Origin	Cali	1.55 (1.14–2.12)	0.005	1.45 (1.01–2.09)	0.040
Adequate prenatal control	Suitable ***	1.70 (1.18–2.44)	0.004	1.46 (0.94–2.27)	0.086
Way of birth	Vaginal	1.68 (1.24–2.27)	0.001	2.29 (1.58–3.33)	<0.001
Necrotizing enterocolitis	Yes	0.16 (0.03–0.79)	0.024	0.18 (0.03–0.97)	0.047
Dysplasia B	Yes	0.04 (0.005–0.308)	0.002	0.13 (0.01–1.13)	0.066
Intraventricular hemorrhage	Yes	0.12 (0.04–0.32)	<0.001	0.15 (0.05–0.45)	0.001
Retinopathy	Yes	0.15 (0.03–0.69)	0.016	0.51 (0.09–2.88)	0.452
Late sepsis	Yes	0.23 (0.16–0.32)	<0.001	0.23 (0.15–0.33)	<0.001

* Odds ratio. ** Multivariate analysis with adjusted OR using logistic regression. *** Adequate prenatal control defined greater than 3 controls.

**Table 6 nutrients-13-02187-t006:** Behavior of the weights of premature infants younger than 37 weeks hospitalized in the neonatal intensive care unit, on admission, 7 days and on discharge.

Weight	Human Milk	Artificial/Mixed	*p*-Value
	*n* = 234	*n* = 217	
Upon admission *	1705	1545	<0.01
On the 7th day *	1697	1570	0.06
Upon discharge *	1910	1930	0.67

* The weight corresponds to the medians of this cohort of babies.

## Data Availability

The data supporting the results reported in this document can be requested from the main researcher’s email: javier.torres@correounivalle.edu.co.

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
