# Peer review of "Clinical Results of the Implementation of a Breast Milk Bank in Premature Infants (under 37 Weeks) at the Hospital Universitario del Valle 2018–2020"

_nutrients, 2021, doi:10.3390/nu13072187_

Round 1
Reviewer 1 Report
Thank you for inviting me to review the manuscript titled " Clinical results of the implementation of a breast milk bank in premature infants (under 37 weeks) at the Hospital Universitario del Valle 2018-2020". The authors in this manuscript describe the results of the implementation of a breast milk bank for preterm infants in a single center in Colombia.
I want to first commend the authors for their work. Indeed, human milk has been shown time and again to be superior to formula at preventing multiple pathologies and improving outcomes. Therefore, exclusive human milk feeds are becoming the standard of care for preterm infants (when available).
The manuscript is relatively well written. I have the following comments/suggestions:
1- In the results section, line 95: The authors mentioned that all preterm infants less than 37 weeks were included in the study except for congenital anomalies and metabolic. Only one infant was excluded here over 2 years. Were there no babies that met the exclusion?
2- In Table one, please explain what HUV means, suitable prenatal controls, change the wording Type to Stage for IVH and ROP. Also please explain how did they define sepsis, and was it early onset or late onset. This is pertinent info as human milk is also known to reduce late-onset sepsis risk. While early-onset sepsis is unrelated to human milk feeds and could affect their outcomes.
3- Also in the table, it seems that 94.88% of infants were intubated, that's a really large percentage given the fact that more than half are above 33 weeks. Can you please elaborate why? what is the unit practice for intubations there? This is important as it's one of the aspects looked at by the authors and affects outcomes such as BPD.
4- In table two, please change enterocolitis to necrotizing enterocolitis and dysplasia to bronchopulmonary dysplasia or use abbreviations if needed.
5- minor issues: change the word taken on line 77 to included. Add percentages to the table next to the numbers in paranthesis.
Author Response
186 / 5000 Dear Dr, we appreciate the suggestions sent during your review, which have been corrected by the authors of the article. We remain attentive to new suggestions or concerns.
Response to Reviewer 1 Comments
Point 1: In the results section, line 95: The authors mentioned that all preterm infants less than 37 weeks were included in the study except for congenital anomalies and metabolic. Only one infant was excluded here over 2 years. Were there no babies that met the exclusion?
Response 1: As regards line 95 of the manuscript, it is that it excluded a patient because it did not have data or records on the variables of interest of the study, such as type of milk received and days. During the study period, there were no patients with congenital malformations or metabolic diseases.
Point 2: In Table one, please explain what HUV means, suitable prenatal controls, change the wording Type to Stage for IVH and ROP. Also please explain how did they define sepsis, and was it early onset or late onset. This is pertinent info as human milk is also known to reduce late-onset sepsis risk. While early-onset sepsis is unrelated to human milk feeds and could affect their outcomes.
Response 2: In line 64 of the manuscript we put the meaning of the acronym, which is the name of the health institution in which the study was carried out (Hospital Universitario del Valle-HUV).
Point 3: Also in the table, it seems that 94.88% of infants were intubated, that's a really large percentage given the fact that more than half are above 33 weeks. Can you please elaborate why? what is the unit practice for intubations there? This is important as it's one of the aspects looked at by the authors and affects outcomes such as BPD.
Response 3: The high percentage in this health institution can be explained by being a regional referral center where 75% of the maternal have significant comorbidities and during intubation those who required it as part of their resuscitation process and those who they remained intubated in the intensive care unit.
Point 4: In table two, please change enterocolitis to necrotizing enterocolitis and dysplasia to bronchopulmonary dysplasia or use abbreviations if needed.
Response 4: The names were modified in the respective tables.
Point 5: Minor issues: change the word taken on line 77 to included. Add percentages to the table next to the numbers in paranthesis.
Response 5: The corrections were made in line 77.
Reviewer 2 Report
The work is interesting, but I think that before its acceptance to be published, the authors must make some changes, like this:
- in section 2.3. lines 76-77, I think the total number of births in that hospital should be added to give an idea of the incidence of prematurity.
-In Table 1, the sum of 722 does not coincide in several sections, for example in the age groups, the sum is 718.
- In the same table, first row of the Scholarship, the number should be placed in figures and they do not add up to 722 at the end.
- This lack of cases occurs in several more sections.
- I think that a fundamental variable should be gestational age, or at least distribute newborns in various subgroups of prematurity.
- In Table 2, heading, I think it should be placed next to maternal (N =) and Other (N =). And then in each absolute frequency, n, in parentheses the relative frequency (%). This would give a better idea of which way the variations are oriented.
- I repeat, both in the bivariate analysis and in the multivariate one, the gestational age should be entered.
- Finally, as a general rule, I think that the value p = 0.000 should not be used and should be replaced by p <0.001.
Author Response
Dear Dr. We appreciate the suggestions sent during your review, which have been corrected by the authors of the article. We remain attentive to new suggestions or concerns.
Point 1: In section 2.3. lines 76-77, I think the total number of births in that hospital should be added to give an idea of the incidence of prematurity.
Response 1: The information on births and prematurity in the study health institution was added on line 77
Point 2: In Table 1, the sum of 722 does not coincide in several sections, for example in the age groups, the sum is 718.
Point 3: In the same table, first row of the Scholarship, the number should be placed in figures and they do not add up to 722 at the end.
Point 4: This lack of cases occurs in several more sections.
Response 2,3,4: Some variables such as the mother's age, schooling, weight-to-height ratio, discharge, route of birth, intraventricular hemorrhage, retinopathy and late sepsis were not kept in all records and were coded as no data, therefore they were not considered in the final tables for its low percentage.
Point 5: I think that a fundamental variable should be gestational age, or at least distribute newborns in various subgroups of prematurity.
Response 5: One of the objectives of this research was to identify the differences in early and late preterm infants, which is why the early preterm and late preterm between 34 and 37 weeks of gestation were classified as under 33 weeks, as shown in Table 1.
Point 6: In Table 2, heading, I think it should be placed next to maternal (N =) and Other (N =). And then in each absolute frequency, n, in parentheses the relative frequency (%). This would give a better idea of which way the variations are oriented.
Response 6: This suggestion was applied in tables 2 and 3 of the manuscript.
Point 7: I repeat, both in the bivariate analysis and in the multivariate one, the gestational age should be entered.
Response 7: The gestational age variable was included in the bivariate and multivariate analysis and those results that were modified with it were corrected. The main findings did not change and therefore the conclusions are the same.
Point 8: Finally, as a general rule, I think that the value p = 0.000 should not be used and should be replaced by p <0.001.
Response 8: Those p values equal to 0.000 was replaced with the term "<0.001" in all the manuscript.